# Cyclic di-GMP Modulates a Metabolic Flux for Carbon Utilization in *Salmonella enterica* Serovar Typhimurium

Jiwon Baek,[a] Hyunjin Yoon[a,b]

[a]Department of Molecular Science and Technology, Ajou University, Suwon, Republic of Korea
[b]Department of Applied Chemistry and Biological Engineering, Ajou University, Suwon, Republic of Korea

**ABSTRACT** *Salmonella enterica* serovar Typhimurium is an enteric pathogen spreading via the fecal-oral route. Transmission across humans, animals, and environmental reservoirs has forced this pathogen to rapidly respond to changing environments and adapt to new environmental conditions. Cyclic di-GMP (c-di-GMP) is a second messenger that controls the transition between planktonic and sessile lifestyles, in response to environmental cues. Our study reveals the potential of c-di-GMP to alter the carbon metabolic pathways in *S.* Typhimurium. Cyclic di-GMP overproduction decreased the transcription of genes that encode components of three phosphoenolpyruvate (PEP):carbohydrate phosphotransferase systems (PTSs) allocated for the uptake of glucose (PTS$^{Glc}$), mannose (PTS$^{Man}$), and fructose (PTS$^{Fru}$). PTS gene downregulation by c-di-GMP was alleviated in the absence of the three regulators, SgrS, Mlc, and Cra, suggesting their intermediary roles between c-di-GMP and PTS regulation. Moreover, Cra was found to bind to the promoters of *ptsG*, *manX*, and *fruB*. In contrast, c-di-GMP increased the transcription of genes important for gluconeogenesis. However, this effect of c-di-GMP in gluconeogenesis disappeared in the absence of Cra, indicating that Cra is a pivotal regulator that coordinates the carbon flux between PTS-mediated sugar uptake and gluconeogenesis, in response to cellular c-di-GMP concentrations. Since gluconeogenesis supplies precursor sugars required for extracellular polysaccharide production, *Salmonella* may exploit c-di-GMP as a dual-purpose signal that rewires carbon flux from glycolysis to gluconeogenesis and promotes biofilm formation using the end products of gluconeogenesis. This study sheds light on a new role for c-di-GMP in modulating carbon flux, to coordinate bacterial behavior in response to hostile environments.

**IMPORTANCE** Cyclic di-GMP is a central signaling molecule that determines the transition between motile and nonmotile lifestyles in many bacteria. It stimulates biofilm formation at high concentrations but leads to biofilm dispersal and planktonic status at low concentrations. This study provides new insights into the role of c-di-GMP in programming carbon metabolic pathways. An increase in c-di-GMP downregulated the expression of PTS genes important for sugar uptake, while simultaneously upregulating the transcription of genes important for bacterial gluconeogenesis. The directly opposing effects of c-di-GMP on sugar metabolism were mediated by Cra (catabolite repressor/activator), a dual transcriptional regulator that modulates the direction of carbon flow. *Salmonella* may potentially harness c-di-GMP to promote its survival and fitness in hostile environments via the coordination of carbon metabolic pathways and the induction of biofilm formation.

**KEYWORDS** *Salmonella* Typhimurium, c-di-GMP, PTS, Cra

Address correspondence to Hyunjin Yoon, yoonh@ajou.ac.kr.

The authors declare no conflict of interest.

*S*almonella enterica serovar Typhimurium (*S.* Typhimurium) is a non-typhoidal serovar of *Salmonella enterica* that causes foodborne illnesses; while generally self-limited, it can occasionally lead to systemic infection and severe inflammation in

10.1128/spectrum.03685-22   **1**

immunocompromised individuals (1). *S.* Typhimurium commonly infects a broad range of animal species and can be shed in feces and transmitted through water and soil (2). During colonization in the preferred hosts and niche adaptation in less favorable environments, *Salmonella* has to sense and respond to changing environments and coordinate the expression of a variety of genes pleiotropically, to aid its environmental fitness. Many bacterial pathogens exploit nucleotide-based second messengers, such as bis-(3′–5′)-cyclic dimeric GMP (cyclic di-GMP; c-di-GMP) and tetra/penta-phosphate guanosine ([p]ppGpp), as key regulatory molecules that orchestrate bacterial behavior, including biofilm formation, cell division, stringent response, quorum sensing, and virulence (3–6). Cyclic di-GMP plays a key role in determining bacterial lifestyles between planktonic (free-living) and sessile (biofilm) states. Cyclic di-GMP metabolic turnover is controlled by multiple diguanylate cyclases and phosphodiesterases. Diguanylate cyclases, which contain GGDEF domains, synthesize c-di-GMP using two GTP molecules, while phosphodiesterases, which contain either EAL or HD-GYP domains, break down one c-di-GMP molecule into one pGpG nucleotide or two GTP molecules, respectively (4, 7). *Salmonella* possesses at least five GGDEF-containing proteins, eight EAL-containing proteins, and seven proteins that contain both the GGDEF and EAL domains (8–11).

Cyclic di-GMP, in cooperation with other global regulators, such as CsgD and $\sigma^S$, promotes intercellular aggregation and adhesion in many bacterial species (12). CsgD is a primary transcriptional regulator that activates the production of curli fimbriae, cellulose, and other exopolysaccharides and the expression of *adrA* encoding a diguanylate cyclase (13–15), and its expression is activated under stress conditions, including low temperature, low osmolarity, and nutrient starvation, where the alternative sigma factor, $\sigma^S$, conducts the general stress response (16, 17). Bacterial cells encased within an extracellular matrix, biofilm, undergo nutrient limitation but also acquire stress tolerance against physical and chemical stimuli (18, 19). The ability to form biofilms is speculated to enable *Salmonella* to withstand unfavorable environments, such as water and soil, while circulating between host species and the environment (20, 21).

Most *Proteobacteria*, including *Salmonella*, utilize the phosphoenolpyruvate (PEP):carbohydrate phosphotransferase system (PTS) to acquire extracellular sugars. PTS comprises two general-purpose cytoplasmic proteins, including enzyme I (EI) and histidine phosphocarrier protein (HPr) and a sugar-specific enzyme II (EII) complex that usually contains two cytosolic domains (EIIA and EIIB) and one transmembrane domain (EIIC), and sometimes an additional transmembrane domain EIID (22, 23). EII complexes, or more specifically, EIIC, spanning the cytoplasmic membrane, differentiate between carbohydrates and transport their cognate sugars, such as glucose, mannose, and fructose, into the cytoplasm by simultaneously phosphorylating them. Two general proteins, EI and HPr, transfer phosphoryl groups from PEP to diverse EII complexes in a cascade. In addition to sugar translocation, the PTS also exerts regulatory functions. Carbon catabolite repression is a well-established regulatory conduit that is controlled by PTS proteins. Enteric bacteria such as *Salmonella enterica* and *Escherichia coli* harness dephosphorylated EIIA$^{Glc}$ to inactivate transporters and metabolic enzymes associated with non-PTS sugars when a preferred sugar, such as glucose, is abundant in the environment (24, 25). In a condition of PTS sugar deficiency, EIIA$^{Glc}$ is more likely to be phosphorylated and subsequently activates adenylate cyclase, leading to an increase in cAMP, which in turn activates the expression of other secondary catabolic genes in combination with cAMP receptor protein (CRP). Recently, an additional regulatory role of EIIA$^{Glc}$ in biofilm formation was revealed in *Vibrio cholerae* (26). In *V. cholerae*, at a high concentration of glucose, dephosphorylated EIIA$^{Glc}$ was shown to strongly bind to PdeS and inactivate its phosphodiesterase activity toward c-di-GMP, thereby leading to the cellular accumulation of c-di-GMP. This accumulation conclusively accelerated biofilm formation and facilitated bacterial colonization and persistence in the host.

This study revealed the possibility of reciprocal regulation between the PTS and c-di-GMP in *S.* Typhimurium. *Salmonella* experiencing a surplus of c-di-GMP downregulated the expression of PTS genes important for the uptake of glucose, mannose, and

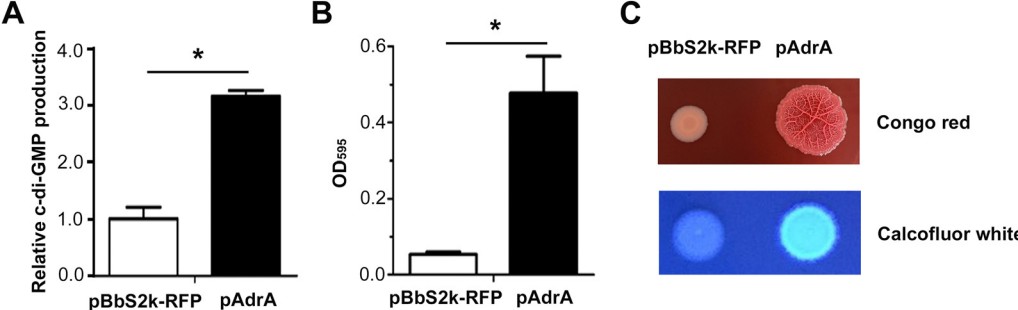

**FIG 1** Overexpression of c-di-GMP stimulated biofilm formation. Wild-type *Salmonella enterica* serovar Typhimurium was transformed with pAdrA or its empty plasmid, pBbS2k-RFP. (A) Cyclic di-GMP assay. Bacterial cells were cultivated in Luria-Bertani (LB) broth containing 10 ng/mL anhydrotetracycline (aTc) for *adrA* induction. (B) Introduction of pAdrA increased biofilm formation. Bacterial strains were cultivated in M9 minimal medium broth containing aTc at 10 ng/mL for 48 h. Biofilm was stained using crystal violet and quantified by measuring the optical density at 595 nm. *, $P < 0.05$. (C) Biofilm induced by pAdrA was analyzed using Congo red and Calcofluor white staining. Bacterial cells were spiked onto LB agar plates containing Congo red (40 $\mu$g/mL) or Coomassie brilliant blue (20 $\mu$g/mL) and incubated at 28°C.

fructose and upregulated the transcription of genes encoding critical components of the gluconeogenesis pathway. We propose that *S.* Typhimurium programmed to form biofilms exploits c-di-GMP for dual purposes, where c-di-GMP stimulates the production of extracellular polymeric substances (EPSs), in cooperation with CsgD, and simultaneously supplements the precursor sugars for EPSs by rewiring carbon metabolic pathways.

## RESULTS

**Cyclic di-GMP downregulated the transcription of PTS$^{Glc}$, PTS$^{Man}$, and PTS$^{Fru}$ genes.** AdrA (or YaiC) is a primary diguanylate cyclase synthesizing c-di-GMP, which is critical for CsgD-mediated cellulose and curli production (27, 28). Overexpression of *adrA* in *S.* Typhimurium increased the cellular c-di-GMP level and stimulated biofilm formation by upregulating the production of curli fimbriae and cellulose (Fig. 1). The *S.* Typhimurium LT2 strain possesses over 20 distinct EII complexes (or PTS permeases) that transport sugar substrates from the extracellular medium through the cytoplasmic membrane. Although not all PTS permeases are conserved among *S.* Typhimurium strains, glucose PTS permease (PtsG/Crr; EIIBC$^{Glc}$/EIIA$^{Glc}$), mannose PTS permease (ManXYZ; EIIABCD$^{Man}$), and fructose PTS permease (FruBA/FrwCB; EIIABC$^{Fru}$/EIICB$^{Fru}$) are prevalent in a variety of *S.* Typhimurium strains whose genome sequences have been reported to date (29–32). *S.* Typhimurium with high c-di-GMP production showed a decrease in the transcription of *ptsG*, *manXYZ*, and *fruBKA*, which encode EIIBC$^{Glc}$, EIIABCD$^{Man}$, and EIIA$^{Fru}$-FPr/fructose-1-phosphate (F1P) kinase/EIIBC$^{Fru}$, respectively (Fig. 2A). Interestingly, *ptsH* and *ptsI*, located farther away from *ptsG* on the chromosome, changed their transcription marginally in response to a high concentration of c-di-GMP. The two proteins encoded by *ptsI* and *ptsH*, EI and HPr, respectively, are universal proteins capable of transferring phosphoryl groups from PEP to diverse EII complexes in a cascade. In contrast, the downregulation of PTS genes by c-di-GMP was alleviated or reversed when RocR, harboring phosphodiesterase activity on c-di-GMP at its C-terminal EAL domain (33) was coexpressed with AdrA (Fig. 2B). In accordance with the downregulation of *ptsG*, *manXYZ*, and *fruBKA*, *S.* Typhimurium with high c-di-GMP production consumed extracellular glucose less than a wild-type *Salmonella* (Fig. 2C).

**Cyclic di-GMP did not influence the expression of regulators associated with PTS regulation.** Toward the search for an intermediary regulator that coordinates the transcription of multiple PTS genes, especially in response to the abundance of c-di-GMP, the transcription of regulators known to regulate PTS genes was investigated when c-di-GMP was synthesized at a high concentration. Bacterial growth curves are presented in Fig. S1. These regulators include transcriptional or posttranscriptional regulatory elements with positive (CRP [34], Fis [35], MtfA or YeeI, [36], CyaA [37], SoxS [38]) or negative (Mlc [39], ArcA [40], Cra or FruR [41, 42], SgrS [43]) effects on the expression of PTS genes. However, none of the regulators showed transcriptional alterations, even at a

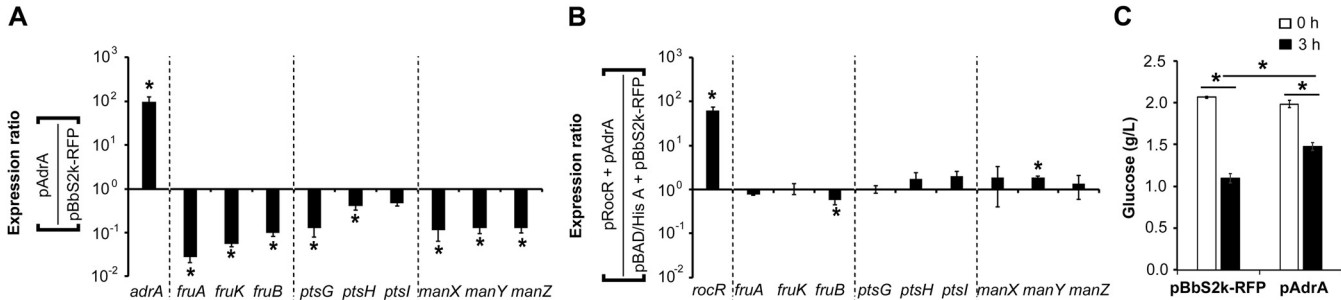

**FIG 2** Cyclic di-GMP downregulates the expression of PTS$^{Glc}$, PTS$^{Man}$, and PTS$^{Fru}$ genes. (A) *S.* Typhimurium harboring pAdrA or pBbS2k-RFP was cultivated in LB broth containing 10 ng/mL aTc for 3 h. Transcription levels of PTS genes were compared between two *S.* Typhimurium strains harboring pAdrA or pBbS2k-FRP using RT-qPCR. The $C_t$ values of each gene were normalized using those of *rpoD* in each strain, and the fold change between two strains (pAdrA/pBbS2k-RFP) was plotted. (B) *Salmonella* harboring pAdrA was transformed with pRocR, while *Salmonella* containing pBbS2k-RFP was transformed with pBAD/His A. Expression of *adrA* and *rocR* was induced by 10 ng/mL aTc and 0.1% arabinose, respectively, for 4 h. Total RNA was isolated from each strain and subjected to RT-qPCR. After normalization of $C_t$ values using those of *rpoD* in each strain, the expression ratio of each gene between two strains (pRocR + pAdrA/pBAD/His A + pBbS2k-RFP) was estimated and plotted. (C) Uptake of glucose was compared between *Salmonella* strains harboring pAdrA or pBbS2k-RFP. Bacterial cells were cultivated in LB broth supplemented with 0.2% glucose and the concentration of extracellular glucose was measured at 3 h. An asterisk indicates *P* value < 0.05.

high concentration of c-di-GMP (Fig. 3). While c-di-GMP did not seem to influence the transcription levels of these regulators, the possibility of c-di-GMP interaction with regulators at posttranscriptional or posttranslational levels, potentially modulating their regulatory activity on PTS genes, cannot be definitively ruled out. To test this possibility, three regulators known to downregulate PTS expression, SgrS, Mlc, and Cra, were further investigated. ArcA, which has a negative role in *ptsG* transcription, was excluded because its regulatory role is proficient in respiratory or fermentative metabolism under anaerobic conditions (44).

**SgrS is required for the downregulation of PTS genes by c-di-GMP.** SgrS is a small RNA (sRNA) that senses cytoplasmic accumulation of phosphorylated sugars, such as glucose-6-phosphate (G6P), and regulates the translation and stability of numerous mRNA targets by base-pairing interactions (45). In response to glucose-phosphate stress, SrgS directly inhibits the translation of *ptsG* and *manXYZ*; it does so by base-pairing with their mRNA transcripts, in cooperation with Hfq, an RNA chaperone protein, and subsequently stimulates RNase E-dependent degradation (46–48). SgrS also encodes SgrT, a 43-amino-acid polypeptide, which serves to selectively block the transport activity of the main glucose permease EIIBC$^{Glc}$ (49). To examine whether SgrS is intercalated into the regulatory circuit between c-di-GMP and PTS gene transcription, the effect of c-di-GMP on PTS regulation was investigated in the absence of SgrS. First, the role of SgrS in the expression of three different PTSs (PTS$^{Glc}$, PTS$^{Man}$, and PTS$^{Fru}$) was examined by introducing pSgrS, which overexpresses SgrS under the control of the P$_{BAD}$ promoter, into the Δ*sgrS* mutant strain (Fig. S2A). SgrS overexpression markedly decreased the levels of *ptsG* and *manXYZ* mRNA transcripts. However, the

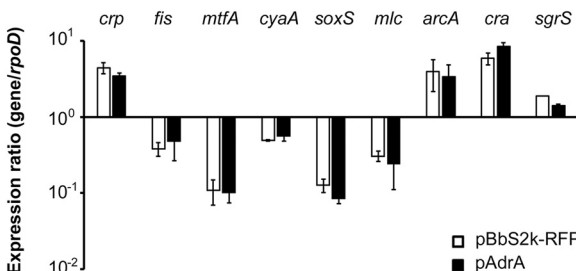

**FIG 3** Cyclic di-GMP overexpression did not substantially alter the expression of regulatory genes associated with PTSs. *S.* Typhimurium strains harboring pAdrA or pBbS2k-RFP were grown in LB broth containing aTc at 10 ng/mL for 3 h. The levels of mRNA were relatively quantified using RT-qPCR. The $C_t$ values of each gene were standardized using those of *rpoD* for each strain; their relative expression ratios are shown.

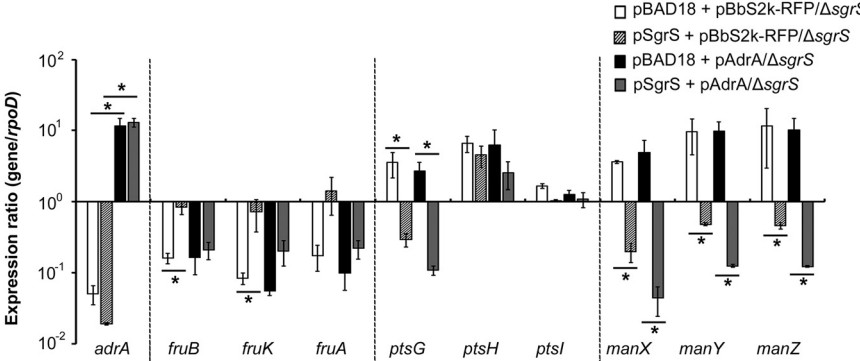

**FIG 4** SgrS is required for the downregulation of PTS genes by c-di-GMP. *S.* Typhimurium Δ*sgrS* strain was transformed with pAdrA or pSgrS, individually or in combination. Plasmids of pBbS2k-RFP and pBAD18, as controls, were introduced to Δ*sgrS* strain in parallel. Bacterial strains were cultivated in M63 minimal medium broth supplemented with 0.2% arabinose for 6 h and then treated with 10 ng/mL aTc for an additional 1 h. RT-qPCR was conducted to estimate transcription levels using $\Delta C_t$ values, where $C_t$ of each gene was subtracted from $C_t$ of *rpoD*. *, $P$ value $< 0.05$.

mRNA levels of the *fruBKA* operon were likely to be increased by SgrS overexpression, which may imply differential regulatory roles of SgrS between PTS$^{Glc/Man}$ and PTS$^{Fru}$. Decreases in the mRNA levels of *ptsG* and *manXYZ* were also observed after the addition of $\alpha$-methylglucoside-6-phosphate ($\alpha$MG), a nonmetabolizing analog of G6P (Fig. S2B), which induces the expression of SrgS and its transcriptional activator SgrR (50). In addition, the negative effect of $\alpha$MG on *ptsG* and *manXYZ* expression was abolished in the absence of Hfq, consistent with previous observations (47). Next, to examine whether c-di-GMP controls the expression of PTS genes via SgrS, c-di-GMP was overproduced in *Salmonella* lacking SgrS using pAdrA and the transcription of PTS genes was evaluated. Notably, in the absence of SgrS, c-di-GMP did not lower transcript levels of PTS$^{Glc}$, PTS$^{Man}$, and PTS$^{Fru}$ genes (Fig. 4), indicating that SgrS mediated the negative effect of c-di-GMP in PTS regulation. Complementation of the *Salmonella* Δ*sgrS* mutant strain with pSgrS restored the negative role of c-di-GMP only in *ptsG* and *manXYZ* expression and not in *fruBKA* operon. In fact, coexpression of SgrS and c-di-GMP resulted in a greater reduction in *ptsG* and *manXYZ* transcript levels than SgrS alone, suggesting that c-di-GMP may promote the negative impacts of SgrS on the regulation of PTS$^{Glc}$ and PTS$^{Man}$. As shown in Fig. S2A, the introduction of pSgrS seemed to increase the transcription levels of *fruBKA*, regardless of a c-di-GMP surplus, implying the possibility of a biphasic SgrS role in *fruBKA* regulation; a positive effect by itself but a negative effect in cooperation with c-di-GMP. Taken together, these results suggest that c-di-GMP requires SgrS to repress PTS$^{Glc}$, PTS$^{Man}$, and PTS$^{Fru}$, whereas SgrS itself may also induce PTS$^{Fru}$ expression independently of c-di-GMP.

**Mlc intervenes in the regulatory circuit between c-di-GMP and PTS genes.** Mlc is a global regulator that controls carbohydrate metabolism and virulence in *Salmonella* (51). In sugar transport systems, Mlc represses the expression of *ptsG* and *manXYZ*, encoding structural components of PTS$^{Glc}$ and PTS$^{Man}$, respectively, and also the *ptsHI* operon, encoding common components of both PTSs (52, 53). On the other hand, the activity of Mlc is reciprocally controlled by EIIB encoded by *ptsG* (54). When abundant glucose is available in the environment, dephosphorylated EIIB sequesters Mlc and blocks its inhibitory role on the PTS. Because of the negative role of Mlc in the transcription of *ptsG* and *manXYZ*, the possibility that c-di-GMP may activate Mlc, thereby leading to the transcriptional downregulation of PTS genes, was examined. Downregulation of PTS$^{Glc}$ and PTS$^{Man}$ genes by c-di-GMP overexpression was nullified in *Salmonella* lacking Mlc (Fig. 5). Interestingly, overexpression of c-di-GMP was still able to partially decrease the transcription levels of *fruBKA* encoding PTS$^{Fru}$, even without Mlc. These results showed that c-di-GMP exerted its negative role in PTS regulation via Mlc, but the influence of

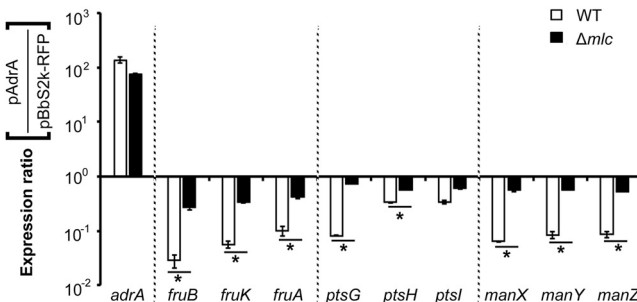

**FIG 5** Mlc mediates the downregulation of PTS genes by c-di-GMP. *S.* Typhimurium WT and Δ*mlc* strains harboring pAdrA or pBbS2k-RFP were cultivated in LB broth containing 10 ng/mL aTc for 3 h. mRNA levels were compared between pAdrA and pBbS2k-RFP in each strain using RT-qPCR, and the expression ratios were plotted. The $C_t$ values of each gene were normalized using those of *rpoD*. *, *P* value $< 0.05$.

Mlc on transmitting the c-di-GMP surplus signal was different between PTS$^{Glc/Man}$ and PTS$^{Fru}$.

**Cra is the pivotal regulator mediating PTS regulation by c-di-GMP.** Cra (catabolite repressor/activator) was initially identified as the fructose repressor, FruR, but was later found to activate or repress the transcription of numerous genes associated with carbon and energy metabolism. Cra is known to repress the *fruBKA* operon and moderately downregulate *pfkA* and *pykF* genes that encode glycolytic enzymes; meanwhile, it also activates *fbp* and *ppsA* genes encoding fructose-1,6-bisphosphatase and PEP synthetase, respectively, which catalyze the two irreversible reactions that differentiate gluconeogenesis from glycolysis (55). The possibility that Cra may coordinate PTS activity in response to c-di-GMP concentration was examined. *Salmonella* overproducing c-di-GMP observably downregulated the expression of multiple genes encoding PTS$^{Glc}$, PTS$^{Man}$, and PTS$^{Fru}$, but deletion of the *cra* gene invalidated the negative role of c-di-GMP in PTS regulation (Fig. 6A). Despite the multifaceted roles of Cra in carbohydrate metabolism, the likelihood that PTS$^{Glc}$ and PTS$^{Man}$ are regulated by Cra has not been substantially elucidated elsewhere. To address this likelihood, the expression of PTS genes was compared between the wild-type and Δ*cra* mutant strains (Fig. 6B). The transcription of *fruBKA* was significantly increased in the absence of Cra, but the introduction of pCra-producing Cra under the P$_{lac}$ promoter suppressed the transcriptional increase of *fruBKA* in the Δ*cra* strain. Surprisingly, the transcription of genes, including *ptsG*, *manXYZ*, and even *ptsHI*, was also increased in the absence of Cra, indicating a negative regulation of PTS$^{Glc}$ and PTS$^{Man}$ by Cra. Their transcriptional increase by *cra* deletion was alleviated with trans-supplementation of Cra, using pCra. These results suggest that Cra might downregulate the activity of the three PTSs in an integrative manner when the cellular c-di-GMP level soars responding to certain environmental cues. To elucidate the mechanism underlying PTS regulation by Cra, the possibility of direct binding of Cra to the promoters of *ptsG* and *manXYZ*, as well as that of *fruBKA*, was examined, both in the presence and absence of fructose-1,6-bisphosphate (FBP) (Fig. 6C). Binding of Cra to its effectors, such as FBP and F1P, dampens its binding affinity toward target promoters, offsetting its role as a transcriptional regulator (56, 57). DNA fragments comprising P$_{fruB}$, P$_{ptsG}$, and P$_{manX}$ were incubated with Cra-His$_6$, and their interactions were determined by staining with ethidium bromide. The bands representing the complexes between Cra and the three promoter regions gradually intensified depending on the concentration of Cra but diminished in the presence of FBP, indicating the direct interaction of Cra with P$_{ptsG}$, P$_{manX}$, and P$_{fruB}$.

**Cra counterbalances sugar transport via PTS and gluconeogenesis in response to c-di-GMP.** Cra is known to play opposing roles in glycolysis and gluconeogenesis. In contrast to its negative effects on the expression of PTS and glycolysis genes, Cra activates the expression of genes critical for gluconeogenesis. To gain insight into the coordinated regulation by c-di-GMP and Cra, the expression of genes associated with

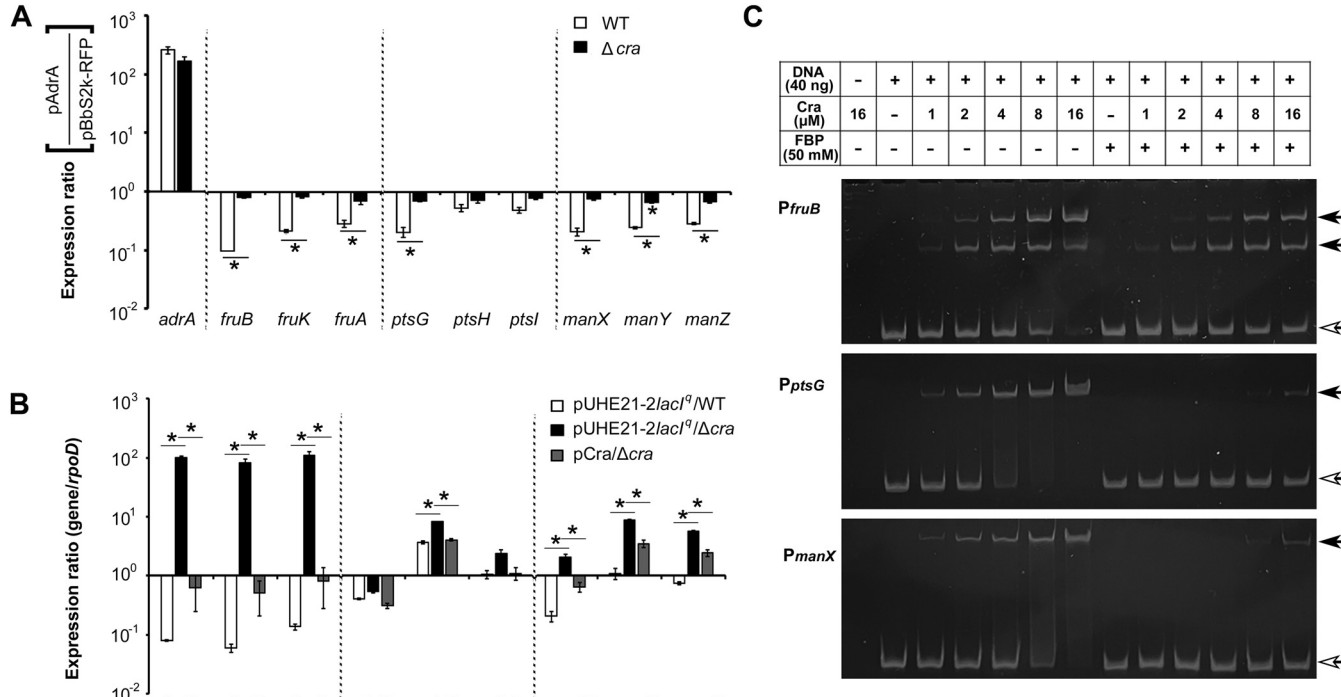

**FIG 6** Cra is the pivotal regulator intercalated between PTS genes and c-di-GMP. (A) *S.* Typhimurium WT and Δ*cra* strains harboring pAdrA or pBbS2k-RFP were cultivated in LB broth containing 10 ng/mL aTc for 5 h. Transcription levels of PTS genes were normalized using those of *rpoD* and compared between pAdrA and pBbS2k-RFP in each strain (WT or Δ*cra*) using RT-qPCR. Expression ratios (pAdrA/pBbS2k-RFP) are depicted. (B) *S.* Typhimurium WT and Δ*cra* strains were transformed with pCra or its empty plasmid, pUHE21-2*lacI^q* and grown in LB broth for 4 h. Isopropyl β-D-1-thiogalactopyranoside (1 mM) was added for *cra* gene induction. The C$_t$ values of each PTS gene were normalized using those of *rpoD* in RT-qPCR and the expression ratios (PTS genes/ *rpoD*) were plotted in each strain. An asterisk indicates a statistically significant difference with *P* value < 0.05. (C) Cra-His$_6$ at different concentrations (0– 16 μM) was incubated with three different DNA fragments containing promoters of *fruBKA* (P$_{fruB}$), *ptsG* (P$_{ptsG}$), and *manXYZ* (P$_{manX}$), in the presence or absence of 50 mM fructose-1,6-bisphosphate (FBP). The complexes between Cra-His$_6$ and DNAs were analyzed using 6% native polyacrylamide gel. Black arrows indicate the DNA-protein complexes, while white arrows are free DNAs of P$_{fruB}$ (237 bp), P$_{ptsG}$ (388 bp), and P$_{manX}$ (331 bp), respectively.

glycolysis, gluconeogenesis, Krebs cycle, and peptidoglycan synthesis was compared in *Salmonella* overexpressing c-di-GMP (Fig. 7). Notably, high concentrations of c-di-GMP increased the transcription of *fbp* (fructose-1,6-bisphosphatase), *ppsA* (PEP synthetase), and *pck* (PEP carboxykinase) genes encoding three essential enzymes for the gluconeogenic pathway and also increased the transcription of *glmS* and *glmU*, which encode enzymes that convert fructose-6-phosphate (F6P) to UDP *N*-acetylglucosamine (UDP-GlcNAc), a precursor of peptidoglycan. In contrast, genes encoding two different isozymes (PfkA/PfkB and PykA/PykF) of the glycolysis pathway showed opposite transcriptional responses to c-di-GMP overexpression between isoforms: increases in *pfkA*

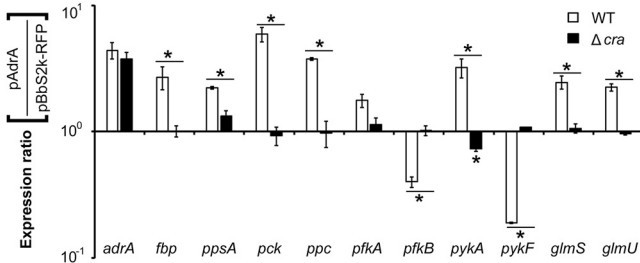

**FIG 7** Genes associated with gluconeogenesis and peptidoglycan synthesis are upregulated by c-di-GMP via Cra. *S.* Typhimurium WT and Δ*cra* strains harboring pAdrA or pBbS2k-RFP were cultivated in LB broth containing 10 ng/mL aTc for 8 h. The levels of mRNA were evaluated using RT-qPCR. The C$_t$ values of each gene were normalized using those of *rpoD*, and the ΔC$_t$ values between pAdrA and pBbS2k-RFP were used to compute the expression ratios (pAdrA/pBbS2k-RFP) in each strain (WT or Δ*cra*). Genes whose transcription was significantly altered by c-di-GMP overexpression are indicated with asterisks (*P* value < 0.05).

and *pykA* transcription, and decreases in *pfkB* and *pykF* transcription. Surprisingly, altered transcription by c-di-GMP overproduction was abolished when the *cra* gene was deleted. These results suggest that c-di-GMP, in combination with Cra, coordinates the direction of carbon flow and consequently determines the metabolic pathways of multiple sugars.

## DISCUSSION

Biofilms are multicellular aggregates enclosed within EPS matrices. Many bacteria determine their multicellular behavior between planktonic (free-living) and sessile (biofilm) modes through a sophisticated regulatory circuit, where c-di-GMP and CsgD orchestrate the transcription of multiple cascades of genes and the activity of their products (9, 58). Cyclic di-GMP is a signaling molecule known to induce bacterial adhesion and intercellular aggregation in a variety of bacteria. More specifically, an increase in the cellular c-di-GMP concentration stimulates the production of cellulose by the *bcsABZC* operon and curli fimbriae by the *csgBAC* operon at the transcriptional or posttranscriptional levels (27). In *Salmonella enterica*, biofilm formation is phenotypically diagnosed by the rdar morphotype, where bacterial aggregative behavior by the production of EPSs, mainly curli fimbriae and cellulose, is characterized by a red color, dryness, and roughness (59). *S.* Typhimurium possesses multiple diguanylate cyclases for c-di-GMP synthesis, but AdrA is a predominant diguanylate cyclase. We observed that *S.* Typhimurium producing an excess of AdrA exhibited the typical rdar morphotype attributable to an increase in curli fimbriae and cellulose production.

Besides biofilm formation, c-di-GMP has also been shown to control bacterial physiological activity in many different ways. For example, a surplus of c-di-GMP attenuates *Salmonella* motility by reducing the production and secretion of flagellin, FliC (9, 60), and compromises *Salmonella* virulence by interfering with the secretion of SopE2, an effector protein required for modulating host cellular cytoskeleton rearrangements during bacterial invasion into host cells (60, 61). Recently, it was found that the biogenesis of c-di-GMP could be controlled by the availability of carbon sources. *V. cholerae* determines whether to stay inside the biofilm or escape from the EPS matrices by modulating the c-di-GMP signaling pathway in response to the availability of preferred sugars, such as glucose (26). When glucose is depleted, phosphorylated EIIA$^{Glc}$ interacts with PdeS, stimulating its phosphodiesterase activity toward c-di-GMP, which in turn triggers EPS deconstruction and biofilm dispersion. In general, the ability to form biofilms is beneficial to many bacterial pathogens for persisting and colonizing under a hostile milieu, because the EPS matrices protect against antimicrobials and the immune response. However, programmed biofilm formation and its subsequent dispersal in response to environmental cues are crucial for bacterial proliferation and dissemination to other sites. In this context, some bacterial pathogens become highly virulent when they spread from multicellular communities, and the dispersed cells are more proficient in transmission to other hosts than planktonic cells (62, 63). Thus, it seems likely that bacteria have evolved dozens of genes responsible for c-di-GMP biogenesis to determine shifts between planktonic and sessile modes in a timely and spatial manner, depending on dynamic environmental cues.

This study revealed that high concentrations of c-di-GMP not only stimulated biofilm formation but also modulated the direction of carbon flow in *Salmonella*. A surplus of c-di-GMP downregulated the transcription of *ptsG*, *manXYZ*, and *fruBKA*, implying that the transport of glucose, mannose, and fructose might be impeded accordingly in *S.* Typhimurium. Notably, *V. cholerae* stimulates c-di-GMP accumulation in response to a high concentration of glucose via the interaction between the EIIA$^{Glc}$ of PTS$^{Glc}$ and PdeS, thereby presumably leading to biofilm formation when glucose is abundant (26). In comparison with *V. cholerae*, where sugars promote biofilm formation, it seems to be discrepant that *S.* Typhimurium dampens sugar transport when biofilm formation is accelerated by c-di-GMP. However, the same signal or stimuli may exert different influences between bacterial species with different host preferences and environmental habitats, as evidenced by numerous studies. For example, Cra represses the transcription of the *fruBKA*

operon in *Salmonella* and *E. coli* but activates the transcription of the same operon by promoting RNA polymerase to bind to the *fru* promoter in *V. cholerae* (64). The downregulation of three PTSs, $PTS^{Glc}$, $PTS^{Man}$, and $PTS^{Fru}$, by c-di-GMP was relieved in whole or in part in the absence of SgrS, Mlc, or Cra, indicating that c-di-GMP indirectly exerts a negative role in PTSs via these three regulators. The negative regulation of *ptsG* and *manXYZ* by SgrS and Mlc has been extensively studied, and it is conceivable that c-di-GMP can manipulate the activity of SgrS and Mlc to downregulate $PTS^{Glc}$ and $PTS^{Man}$. This study revealed a central role of Cra in the regulation of three different PTSs by c-di-GMP. In contrast to SgrS and Mlc, whose absence moderately abrogated downregulation by c-di-GMP, the lack of Cra fully suppressed the negative influence of c-di-GMP on the three PTSs. The consensus Cra box (65, 66) was predicted in the sequences upstream of *ptsG* and *manXYZ* as well as *fruBKA* (Fig. S3). We found that Cra could directly bind to the promoters of *fruBKA*, *ptsG*, and *manXYZ* containing the consensus Cra boxes and repress their transcription. Taken together, we speculate that c-di-GMP exploits Cra primarily for coordinated downregulation of the three PTSs. Interestingly, the transcription levels of the three intermediary regulators, SgrS, Mlc, and Cra, were not altered by a surplus of c-di-GMP, suggesting that c-di-GMP may regulate them at translation or posttranslational events. With regard to posttranslational regulation, c-di-GMP binding to a target protein can modulate the activity of the partner protein by altering its structural conformation and influencing its interaction with others, as demonstrated in PilZ domain proteins (67). A variety of proteins involved in bacterial sessility, motility, and virulence have been proposed to interact with c-di-GMP (68). As a new role for c-di-GMP in controlling carbon metabolic pathways, it is valuable to further test whether c-di-GMP binds to Mlc and Cra and modulates their binding affinities toward the promoters of PTS genes. The binding affinity of Cra to the promoters of *fruBKA*, *ptsG*, and *manXYZ* was analyzed in the presence and absence of c-di-GMP (Fig. S4) as an effort to define how c-di-GMP modulates the regulatory activity of Cra (Fig. S4). Even with the inhibitory FBP, the addition of c-di-GMP tended to facilitate Cra to bind to $P_{fruB}$ and $P_{manX}$. However, c-di-GMP failed to restrain FBP which attenuated the binding affinity of Cra toward $P_{ptsG}$, indicating differential binding affinities between promoters.

AdrA as a membrane-associated protein is expected to have a C-terminal GGDEF domain and an N-terminal MASE2 (membrane-associated sensor 2) domain. The MASE2 domain has been postulated to perceive cellular signals and is frequently seen paired with the catalytic domains in diguanylate cyclases and adenylate cyclases (69). However, the signal recognized by MASE2 domain is unknown. We looked into the possibility that AdrA can respond to the metabolic intermediates associated with SgrS, Mlc, and Cra and consequently modulate the production of c-di-GMP. The absence of either regulator had no effect on the cellular concentrations of c-di-GMP (Fig. S5A), indicating that these regulators did not influence the activity of the diguanylate cyclase, AdrA. When the morphotypes were compared between bacterial strains, however, the absence of SgrS, Mlc, or Cra promoted the development of the rdar and rugose morphotypes, which is predominantly attributed to the activity of c-di-GMP (Fig. S5B). We presume that these regulators may obstruct the function of c-di-GMP or that they may regulate the production of curli and cellulose independently of c-di-GMP.

In contrast to PTS genes, c-di-GMP increased the expression of genes associated with gluconeogenesis, and the positive regulation by c-di-GMP was also mediated by Cra. Considering that the primary role of c-di-GMP is to promote biofilm formation, it is plausible that c-di-GMP may rewire carbon flux to gluconeogenesis, which supplies F6P and glucosamine-6-phosphate (GlcN-6P), the starting substrates of structural polysaccharides, such as peptidoglycan and various extracellular polysaccharides (70, 71). In accordance with the increased expression of *fbp*, *ppsA*, and *pck* genes encoding three critical enzymes for gluconeogenesis at a high concentration of c-di-GMP, two genes, *glmS* and *glmU*, required for cell wall biosynthesis, also showed transcriptional increases in response to a surplus of c-di-GMP. UDP-GlcNAc serves as a precursor of cell wall components and is synthesized from F6P through four enzymatic reactions

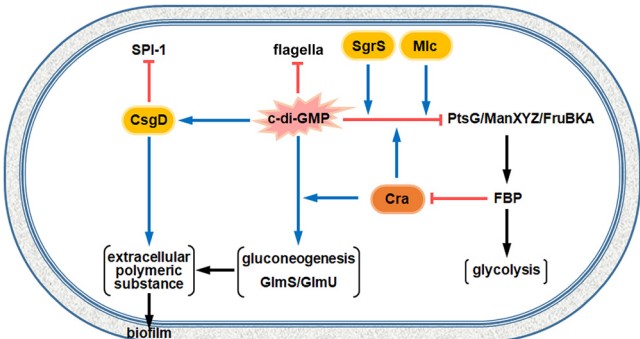

**FIG 8** A schematic regulatory model of c-di-GMP-mediated coordination between biofilm formation and carbon metabolism in *S.* Typhimurium. Besides aiding in biofilm formation, c-di-GMP, in cooperation with SgrS, Mlc, and Cra, also downregulates the expression of three dominant PTSs (PTS$^{Glc}$, PTS$^{Man}$, and PTS$^{Fru}$), while simultaneously upregulating the expression of genes associated with gluconeogenesis and amino sugar synthesis, which can substantially supply the substrates for biofilm. Cyclic di-GMP is also known to attenuate the bacterial motility and virulence attributable to *Salmonella* pathogenicity island (SPI)-1. Blue arrows and red blunt arrows indicate positive and negative regulation, respectively. Oval symbols are regulatory RNA or proteins. Black arrows indicate the direction of carbon flux.

catalyzed by GlcN-6P synthase (GlmS), phosphoglucosamine mutase (GlmM), and bifunctional enzyme glucosamine-1-phosphate acetyltransferase/*N-acetylglucosamine*-1-phosphate uridyltransferase (GlmU) (72). Considering that the three PTSs of PTS$^{Glc}$, PTS$^{Man}$, and PTS$^{Fru}$ consume PEP in common as a phosphate donor for their intermediate phosphotransferases, operating these PTSs in excess may lead to the exhaustion of PEP resources for the synthesis of structural polysaccharides constituting multicellular matrices. To promote bacterial fitness under unfavorable conditions, *Salmonella* may harness c-di-GMP as a dual-purpose signaling molecule, which stimulates biofilm formation in concert with CsgD and coordinates carbon metabolism to supply substrates for biofilm by downregulating PTSs activity while upregulating gluconeogenesis and amino sugar synthesis (Fig. 8).

## MATERIALS AND METHODS

**Bacterial strains, plasmids, and growth conditions.** *Salmonella enterica* subsp. *enterica* serovar Typhimurium 14028S (ATCC 14028S) was used as the wild-type strain. *S.* Typhimurium 14028S strains lacking *hfq* (73), *mlc* (51), or *cra* (73) were constructed using the phage λ-derived Red recombination system (74), as described in our previous studies. *S.* Typhimurium Δ*sgrS* was also constructed using the phage λ Red recombination system. Briefly, the kanamycin resistance (*kan*) cassette of pKD13 was PCR-amplified using two primers (srgS-RF and sgrS-RR, Table S1) designed to provide 40-nucleotide sequences adjacent to *sgrS* at both termini of the resultant PCR products. The PCR products were introduced into *Salmonella* harboring pKD46 (74), to replace *sgrS* with a *kan* cassette. The *kan* marker was subsequently removed using pCP20 (74) providing a flip recombinase. The *mlc::kan* allele of *S.* Typhimurium SL1344 (51) was transferred to *S.* Typhimurium 14028S using P22 HT105/1 int-201-mediated transduction (75) and the *kan* marker was removed using pCP20.

The plasmid pAdrA was constructed using pBbS2K-RFP (76). The *adrA* gene was PCR-amplified using primers pAdrA-CF and pAdrA-CR, and the PCR product was inserted into pBbS2K-RFP through BglII and XhoI sites. For pSgrS plasmid construction, DNA sequences encompassing *srgS* were synthesized as described previously (77) and cloned via the AgeI and HindIII sites of pBAD18 (78). pRocR (kindly gifted by Jang Won Yoon from Kangwon National University, Republic of Korea) was constructed by cloning the *Pseudomonas aeruginosa rocR* gene into pBAD/His A (Invitrogen$^T$) using SacI and HindIII. To construct pCra, the *cra* gene was amplified by PCR using primers pCra-CF and pCra-CR, treated with SalI and BamHI, and then cloned into pUHE21-2*lacI^q* (79). To express Cra tagged with His$_6$ at its C terminus, the *cra* gene was PCR-amplified using the primers Cra-6×His-F and Cra-6×His-R and inserted into the pBAD33 (78) plasmid through SacI and XbaI. Primers used for plasmid construction are listed in Table S1.

Unless otherwise noted, bacterial strains were grown in Luria-Bertani (LB) medium at 37°C. When minimal medium broth was used, bacterial cells were precultured in LB overnight and then diluted 1:100 in fresh M9 or M63 minimal medium broths. Antibiotics and inducers were added at the following concentrations: kanamycin, 50 μg/mL; ampicillin, 50 μg/mL; chloramphenicol, 35 μg/mL; anhydrotetracycline (aTc), 10 ng/mL; arabinose, 0.1% or 0.2%; isopropyl β-D-1-thiogalactopyranoside (IPTG), 1 mM. Methyl α-d-glucopyranoside (αMG) was used at a concentration of 0.5%. All the chemical reagents were purchased from Sigma-Aldrich.

**Cyclic di-GMP assay.** Cyclic di-GMP was extracted using the heat and ethanol precipitation method (80). Bacterial cells were treated with a 0.19% formaldehyde solution on ice for 10 min and washed with distilled

water. Cells resuspended in distilled water were heated at 100°C for 5 min and mixed with ethanol at a 3:7 ratio. Bacterial cells were homogenized using a 20-gauge needle on ice and centrifuged at 17,000 $\times$ $g$ for 10 min. The supernatant was dried using a 1 L rotary evaporator (EYELA) and frozen at $-80$°C. The frozen sample containing c-di-GMP was lyophilized using a TFD5503 Bench-Top freeze-dryer (ilShin Biobase Europe) and analyzed using a c-di-GMP ELISA kit (EIAab), according to the manufacturer's instructions.

**Biofilm assay.** Bacterial cells were cultivated statically in glass tubes at 37°C for 48 h. The glass tubes were emptied and washed twice to remove free-living bacteria. Biofilms attached to glass surfaces were stained with 1% crystal violet (Sigma-Aldrich) for 15 min. The dye absorbed in the biofilm structure was released using 33% acetic acid and was measured at $OD_{595}$.

**Congo red-based rdar morphotyping and Calcofluor white staining.** LB agar plates depleted of NaCl were supplemented with Congo red (40 $\mu$g/mL, Sigma-Aldrich) and Coomassie brilliant blue (20 $\mu$g/mL, Sigma-Aldrich) for rdar morphotyping or with Calcofluor white M2R (10 $\mu$g/mL, Sigma-Aldrich) for Calcofluor white staining. Equivalent numbers of bacterial cells ($OD_{600}$ = 1.0) were spotted onto the modified LB agar plates and incubated at 28°C. For Calcofluor white staining, agar plates were observed under UV light (VILBER UV Hand Lamp BVL-8; Lab Unlimited).

**Quantitative reverse transcription-PCR (RT-qPCR) analysis.** Bacterial cells were treated with RNAprotect Bacteria Reagent (Qiagen) to retard RNA degradation and subsequently subjected to total RNA extraction using an RNeasy minikit (Qiagen). The isolated total RNA was treated with RNase-free DNase (Ambion) at 37°C for 30 min to remove genomic DNA contaminants. cDNA was synthesized using RNA to cDNA EcoDry Premix (TaKaRa Bio, Inc.), and an aliquot corresponding to 10 ng of input RNA was used as a template in each RT-qPCR. PCR primers (Table S2) were designed to produce amplicons of 100 to 150 bp in size, and the levels of amplified PCR products were normalized to those of *rpoD*, a housekeeping gene. The StepOnePlus real-time PCR system (Applied Biosystems) was used for RT-qPCR with THUNDERBIRD SYBR qPCR Master Mix (Toyobo).

**Quantification of glucose in aqueous solution.** Bacterial cells were cultivated in LB broth supplemented with 0.2% glucose. The culture medium was sampled every hour and the concentration of glucose in the supernatant was determined using the Cedex Bio Analyzer (Roche).

**Purification of His$_6$–tagged Cra.** *Escherichia coli* harboring pCra-His$_6$ was cultivated in LB broth and treated with 0.2% arabinose for 3 h for Cra-His$_6$ induction (81). Bacterial cells were harvested at 10,000 $\times$ $g$ for 10 min and treated with 1 mg/mL lysozyme in lysis buffer (50 mM NaH$_2$PO$_4$, 300 mM NaCl, and 20 mM imidazole, pH 8.0) on ice for 30 min. The cell suspension was sonicated and centrifuged at 10,000 $\times$ $g$ at 4°C for 20 min. The soluble lysate fraction was then mixed with Ni$^{2+}$-nitrilotriacetic acid (Ni$^{2+}$-NTA) agarose beads (Qiagen) and incubated at 4°C for 1 h with gentle agitation. The reactant was loaded onto a Ni$^{2+}$-NTA agarose affinity column (Qiagen). The column was washed with washing buffer (50 mM NaH$_2$PO$_4$, 300 mM NaCl, and 40 mM imidazole, pH 8.0) three times, and Cra-His$_6$ bound to the column was eluted using elution buffer (50 mM NaH$_2$PO$_4$, 300 mM NaCl, and 300 mM imidazole, pH 8.0). The eluted fraction was placed in a dialysis membrane tubing with 10 K MWCO (SnakeSkin Dialysis tubing, Thermo Scientific Inc.) and incubated in dialysis buffer (20 mM Tris-HCl, pH 8.0, 150 mM NaCl, 0.1 mM EDTA, 5 mM DTT, and 20% glycerol) at 4°C. Protein concentration was measured using the Bradford assay (Bio-Rad) (82).

**Electrophoretic mobility shift assay (EMSA).** The binding of Cra-His$_6$ to P$_{fruB}$, P$_{ptsG}$, and P$_{manX}$ was examined using EMSA, as described previously (81, 83, 84), with slight modifications. DNA fragments comprising the promoter regions of *fruB*, *ptsG*, and *manX* were PCR-amplified using the primers listed in Table S2 and purified using the Wizard SV Gel and PCR Clean-Up System (Promega). DNA (40 ng) was incubated with different concentrations of Cra-His$_6$ (0, 1, 2, 4, 8, and 16 $\mu$M) in a binding buffer (10 mM Tris-HCl, pH 8.0, 5% [vol/vol] glycerol, 0.1 mM EDTA, and 1 mM DTT) at 37°C for 20 min. Fructose-1, 6-bisphosphate (FBP; F6803, Sigma-Aldrich) was added at different concentrations (0 to 50 mM) as a competitor to block the interaction between Cra-His$_6$ and DNAs (85). Cyclic di-GMP (SML1228, Sigma-Aldrich) was added at 250 $\mu$M, when required. The reaction was analyzed by electrophoresis on a 6% native polyacrylamide gel. The gel was stained with ethidium bromide solution, and DNA fragments were observed using a ChemiDoc MP System (Bio-Rad).

**Statistical analysis.** The experiments were repeated at least three times. For quantitative evaluation, including the c-di-GMP assay, glucose assay, biofilm assay, and RT-qPCR, the average values were computed from at least three independent tests and presented with their standard deviations. Statistical analysis was conducted using Student's *t* test and one-way analysis of variance (ANOVA) with Tukey's *post hoc* test. All statistical analysis employed GraphPad Prism 5 (GraphPad Software Inc., CA, USA). The difference was considered to be significant at *P* value $< 0.05$.

## SUPPLEMENTAL MATERIAL

Supplemental material is available online only.
**SUPPLEMENTAL FILE 1**, PDF file, 0.6 MB.

## ACKNOWLEDGMENTS

We thank Jang Won Yoon (Kangwon National University, Republic of Korea) for providing pRocR. This study was supported by the Bio & Medical Technology Development Program of the National Research Foundation of Korea (NRF) funded by the Ministry of Science & ICT (2021M3A9I4026029).

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
