## [Reviewer comments · Microbiology Spectrum]

Microbiology Spectrum

Cyclic di-GMP modulates a metabolic flux for carbon utilization in *Salmonella enterica* serovar Typhimurium

Hyunjin Yoon and Jiwon Baek

Corresponding Author(s): Hyunjin Yoon, Ajou University

Review Timeline:

Submission Date:	September 12, 2022
Editorial Decision:	October 24, 2022
Revision Received:	January 10, 2023
Accepted:	January 16, 2023

Editor: Cristina Solano

Reviewer(s): Disclosure of reviewer identity is with reference to reviewer comments included in decision letter(s). The following individuals involved in review of your submission have agreed to reveal their identity: Victor H. Bustamante (Reviewer #2)

Transaction Report:

DOI: <https://doi.org/10.1128/spectrum.03685-22>

October 24, 2022

Dr. Hyunjin Yoon
Ajou University
Suwon
Korea (South), Republic of

Re: Spectrum03685-22 (Cyclic di-GMP modulates a metabolic flux for carbon utilization in *Salmonella enterica* serovar Typhimurium)

Dear Dr. Hyunjin Yoon:

Thank you for submitting your manuscript to Microbiology Spectrum. Your work has been revised by two experts in the field who consider the topic to be interesting, though some modifications need to be addressed to strengthen the conclusions of the manuscript. As it is, with only gene expression studies, the model presented is still premature. Their comments are provided at the bottom of this letter. Taking these comments into account, I wish to consider a revised version of the manuscript that adequately addresses all criticisms and concerns expressed by both reviewers.

Link Not Available

Sincerely,

Cristina Solano

Journals Department
Reviewer comments:

Reviewer #1 (Comments for the Author):

This manuscript by Baek and Yoon explores the role of cyclic di-GMP (cdG) in the central metabolism of *S. Typhimurium*. The authors find using gene expression studies that cdG downregulates PTS components while upregulating gluconeogenesis pathways genes. They suggest the regulators SgrS (sRNA), Mlc, and Cra are necessary for this response, with Cra having the

most direct effect. However, how these regulators mediate this response is not determined. Showing an effect of cdG on central metabolism is interesting, and I find their model that cdG helps to switch *S. Typhimurium* to a metabolic state more favorable for extracellular polysaccharide production something that would be of interest to the field. However, the manuscript does not provide any evidence that the measured gene changes have any functional consequences, which would greatly strengthen its conclusions. Furthermore, the authors need to explore if mutations in the central regulators impact basal cdG or the activity of the diguanylate cyclase, AdrA, which they use to alter cdG levels.

1. Line 74-make this a new paragraph
2. Line 102, 294-change "cholera" to "cholerae"
3. In some species, overproduction of cdG can slow growth. The authors should perform a growth curve of their WT and AdrA overexpressing strain to test if this is a factor in their experiments.
4. Line 159-It would be prudent to mention that SgrS also encodes the SgrT peptide, which inhibits PstG transport of glucose.
5. Fig. 4-Lines 175-177. It is not possible to determine if the reduction in ptsB and manXYZ is due to cdG or SrgS overexpression because this experiment is missing one key strain, the pSrgS + pBbS2k (vector control) to show what effect overproduction of SgrS has on expression of these genes in the absence of cdG overproduction from AdrA. Without this strain, it is not possible to appreciate how much of the reduced expression in the SgrS/AdrA strain is due to cdG or SgrS (NOTE: I realize they explored SgrS alone in Fig. S1, but this did not have the other vector control and was not done side by side with AdrA expression).
6. Although I do not find this likely, it is possible that mutations in the metabolic regulators SgrS, Mlc, and Cra could influence the diguanylate cyclase activity of AdrA. The reason this is a possibility is that most DGCs have N-terminal sensory domains that respond to either environmental or host derived cues, and AdrA could be sensing metabolic intermediates in the cell. Such regulation of AdrA activity could explain many of their results. The authors should therefore quantify cdG upon overexpression of AdrA in these transcription factor mutants to test this possibility. In addition, mutations to these key regulators could impact the basal concentrations of c-di-GMP even in the absence of AdrA which may impact their expression studies.
7. Lines 193-195-This statement is speculation without evidence. There are many possible mechanisms. It should be removed.
8. Fig. 6C-This EMSA experiment with Cra should be tested with the addition of cdG to determine if cdG directly impacts binding of Cra to its target promoters. Given that the EMSA has already been developed, that is a simple experiment to try which greatly informs the model in Fig. 8.
9. One weakness of the paper is that all the conclusions are based on gene expression results. It would strengthen the paper to demonstrate a functional consequence to the demonstrated cdG gene regulation. Are there specific growth conditions or substrates that can be used to demonstrate that cdG is decreasing PTS activity while increasing gluconeogenesis?
10. The authors use standard T-tests to analyze statistical differences in large datasets, but they should really be using ANOVAs.

Reviewer #2 (Comments for the Author):

This manuscript shows a new role for c-di-GMP, which is the modulation of carbon flux. The authors analyzed the effect of the overproduction of c-di-GMP in *S. Typhimurium*, by overexpressing the diguanylate cyclase AdrA, and found that it decreased the expression of genes encoding components of phosphoenolpyruvate (PEP):carbohydrate phosphotransferase systems (PTSs) for the uptake of glucose, mannose and fructose. Furthermore, c-di-GMP was shown to act through the regulators SrgS, M1c and Cra to control the three PTSs, being Cra the major factor for this c-di-GMP-mediated regulation. The authors demonstrate that Cra binds to genes of the three PTSs. Additionally, they found that c-di-GMP-Cra also regulates the expression of genes for gluconeogenesis but positively. Thus, this manuscript supports that c-di-GMP rewires carbon flux from glycolysis to gluconeogenesis to favor biofilm formation, which requires the end products of gluconeogenesis. The manuscript is well written, and conclusions are completely supported by data.

Comments.

What about deletion of adrA? Does it affect the expression of genes for carbon flow?

It would be good to mention in Discussion that results from EMSAs (number of DNA/protein complexes) agree with the number of predicted Cra-binding sites on fruB, pstG and manX.

Do the genes for gluconeogenesis also harbor the CsrA-binding consensus sequence?

Minor comments.

Line 218. pCra instead pFruR

Line 226. The bands from EMSAs were seen by staining with ethidium bromide, not by using an anti-His antibody.

Staff Comments:

Preparing Revision Guidelines

Please return the manuscript within 60 days; if you cannot complete the modification within this time period, please contact me. If you do not wish to modify the manuscript and prefer to submit it to another journal, please notify me of your decision immediately so that the manuscript may be formally withdrawn from consideration by Microbiology Spectrum.

January 10, 2023

Manuscript Number: Spectrum03685-22

Dear Editor,

We appreciate the faithful and valuable review. Our detailed responses to the review comments are provided below and the changes are highlighted in blue in the revised manuscript. We have addressed all of the concerns of the reviewers and hope that the revised manuscript is now suitable for considering publication in *Microbiology Spectrum*.

Reviewer comments:

Reviewer #1 (Comments for the Author):

This manuscript by Baek and Yoon explores the role of cyclic di-GMP (cdG) in the central metabolism of *S. Typhimurium*. The authors find using gene expression studies that cdG downregulates PTS components while upregulating gluconeogenesis pathways genes. They suggest the regulators SgrS (sRNA), Mlc, and Cra are necessary for this response, with Cra having the most direct effect. However, how these regulators mediate this response is not determined. Showing an effect of cdG on central metabolism is interesting, and I find their model that cdG helps to switch *S. Typhimurium* to a metabolic state more favorable for extracellular polysaccharide production something that would be of interest to the field. However, the manuscript does not provide any evidence that the measured gene changes have any functional consequences, which would greatly strengthen its conclusions. Furthermore, the authors need to explore if mutations in the central regulators impact basal cdG or the activity of the diguanylate cyclase, AdrA, which they use to alter cdG levels.

Response: We value the reviewer's perceptive feedback. We acknowledge that the experimental evidence based just on gene regulation is insufficient to predict the functional outcome. Therefore, in the revised manuscript, we further explored whether the downregulation of three PTSs by c-di-GMP might slow the pace of extracellular glucose import. We also looked into the possibility that the absence of three regulatory mediators-SgrS, Mlc, and Cra-could influence the production of c-di-GMP and its cellular activity. The detailed results are discussed in the relevant comments below.

1. Line 74-make this a new paragraph

Response: According to the reviewer's suggestion, the sentences describing the roles of c-di-GMP, CsgD and σ^S in biofilm formation and stress adaptation have been separated, making a new paragraph (line 75 - 85 in the revised).

2. Line 102, 294-change "cholera" to "cholerae"

Response: We appreciate the considerate correction. Bacterial name has been corrected according to the binomial nomenclature (lines 103 and 301).

3. In some species, overproduction of cdG can slow growth. The authors should perform a growth curve of their WT and AdrA overexpressing strain to test if this is a factor in their experiments.

Response: We were also aware of the growth deficiency triggered by the overexpression of c-

c-di-GMP, as the reviewer mentioned. To minimize the growth inhibitory effect of c-di-GMP, we optimized the induction condition by employing multiple strategies, including a tight regulation promoter, a low copy number plasmid, and low concentrations of inducer. Comparison of bacterial growth curves is provided in Fig. S1 in the revised manuscript. Despite several attempts, *Salmonella* producing c-di-GMP showed attenuated growth, reaching an OD₆₀₀ of 2.3 at its maximum, which was around 30% lower than the control with an empty plasmid. Instead, we used bacterial cells that were in the same stage of growth. The PTS genes were tested at the log phase, or approximately 3 h in both strains, while the gluconeogenesis-associated genes were examined at the stationary phase, which was roughly 8 h in both cases. The poor growth caused by c-di-GMP overexpression, which downregulates the expression of three PTSs, may be an inevitable result given that the three PTSs are primarily in charge of acquiring carbon sources.

4. Line 159-It would be prudent to mention that SgrS also encodes the SgrT peptide, which inhibits PstG transport of glucose.

Response: We appreciate the thoughtful suggestion. The role of SgrS encoding SgrT has been added in the revised text with a relevant reference (line 163 – 165).

5. Fig. 4-Lines 175-177. It is not possible to determine if the reduction in ptsB and manXYZ is due to cdG or SrgS overexpression because this experiment is missing one key strain, the pSrgS + pBbs2k (vector control) to show what effect overproduction of SgrS has on

expression of these genes in the absence of cdG overproduction from AdrA. Without this strain, it is not possible to appreciate how much of the reduced expression in the SgrS/AdrA strain is due to cdG or SgrS (NOTE: I realize they explored SgrS alone in Fig. S1, but this did not have the other vector control and was not done side by side with AdrA expression).

Response: We concur with the reviewer that the previous version of Fig. 4 was insufficient to distinguish the effects between SgrS and c-di-GMP. Therefore, the revised Fig 4 has been made using appropriate strains. The $\Delta sgrS$ mutant strain was transformed with pSgrS or pAdrA, individually or in combination and the relevant empty plasmids (pBAD18 and pBbs2k-RFP) were also introduced in parallel. In the side-by-side comparison, c-di-GMP failed to downregulate the expression of PTS genes in the absence of SgrS, which was consistent with the earlier finding that SgrS was a mediator of c-di-GMP's influence on PTS regulation. Interestingly, SgrS alone could exert a negative effect on the expression of PTS^{Glc} and PTS^{Man} genes, as expected, and the co-expression of SgrS and c-di-GMP synergistically downregulated their mRNA levels. This result suggests that c-di-GMP may facilitate the negative role of SgrS in the regulation of PTS^{Glc} and PTS^{Man} . Meanwhile, SgrS alone activated the expression of PTS^{Fru} , which was accordant to the previous result (Fig. S2 in the revision). The revised manuscript has been supplemented with the revised Fig. 4 and the relevant description (line 182 - 185).

6. Although I do not find this likely, it is possible that mutations in the metabolic regulators SgrS, Mlc, and Cra could influence the diguanylate cyclase activity of AdrA. The reason this is a possibility is that most DGCs have N-terminal sensory domains that respond to either environmental or host derived cues, and AdrA could be sensing metabolic intermediates in

the cell. Such regulation of AdrA activity could explain many of their results. The authors should therefore quantify cdG upon overexpression of AdrA in these transcription factor mutants to test this possibility. In addition, mutations to these key regulators could impact the basal concentrations of c-di-GMP even in the absence of AdrA which may impact their expression studies.

Response: Sequence analysis based on the Pfam database predicts that AdrA is a membrane associated protein with an N-terminal MASE2 domain and a C-terminal GGDEF domain. The MASE2 domain contains four predicted transmembrane helices and is proposed to sense cellular signals. However, the signal recognized by MASE2 domain is unknown, although this domain is often found associated with nucleotide cycling domains in diguanylate cyclases and adenylate cyclases (Topal, H. et al. 2012. *J Mol Biol* 416, 271-286). To determine whether AdrA responds to the metabolic intermediates associated with SgrS, Mlc, and Cra and consequently controls the production of c-di-GMP, the levels of c-di-GMP were compared between a wild-type and three mutant strains lacking SgrS, Mlc, or Cra. To ensure the production of AdrA similar amongst these strains, pAdrA was used to transform all strains. Under the tested condition (LB broth, 37°C, 48 h), the absence of either regulator had no effect on the cellular concentrations of c-di-GMP (Fig. S5A in the revised manuscript), indicating that the absence of these regulators did not influence the activity of the diguanylate cyclase, AdrA. The primary role of c-di-GMP is to stimulate biofilm formation by upregulating the production of curli and cellulose, both of which may be easily recognized by rdar and rugose morphotypes. When the morphotypes were compared between bacterial strains, the mutants lacking SgrS, Mlc, or Cra promoted the development of the rdar and rugose morphotypes, which was unexpected because the c-di-GMP

levels were comparable between the wild-type and these mutant strains. This result suggests that these regulators may obstruct c-di-GMP action or independent of c-di-GMP, they may also regulate the production of curli and cellulose. This result has been added as Fig. S5B in the revision. In addition, we also examined the possibility that these three regulators could impact the basal concentrations of c-di-GMP even in the absence of AdrA. Unfortunately, the basal c-di-GMP levels were too low to compare between the wild-type and three mutant strains. Instead, we compared the rdar morphotypes between bacterial strains without AdrA overexpression (see bacterial strains harboring pBbS2k-RFP, an empty plasmid in Fig. S5B). Bacterial strains did not show different morphotypes at 37°C. However, at 28°C, the Δmlc and Δcra strains suppressed the rdar morphotype formation, indicating the positive roles of Mlc and Cra in the production of curli and cellulose. Due to the opposite roles of these regulators in biofilm formation depending on the presence of pAdrA, we have opted not to interpret this result in the revision but want to look into the roles of these regulators in the expression of the curli and cellulose genes. Instead, in Discussion, we proposed the possibility that these regulators might modulate bacterial behavior attributable to the activity of c-di-GMP (line 328 - 340).

7. Lines 193-195-This statement is speculation without evidence. There are many possible mechanisms. It should be removed.

Response: The statement has been removed according to the reviewer's suggestion.

8. Fig. 6C-This EMSA experiment with Cra should be tested with the addition of cdG to determine if cdG directly impacts binding of Cra to its target promoters. Given that the EMSA has already been developed, that is a simple experiment to try which greatly informs the model in Fig. 8.

Response: According to the suggestion, the binding affinity of Cra to the promoters of *fruBKA*, *ptsG*, and *manXYZ* were examined in the presence of c-di-GMP. Even with the inhibitory FBP, the addition of c-di-GMP tended to facilitate Cra to bind to P_{fruB} and P_{manX} . However, c-di-GMP failed to restrain FBP which attenuated the binding affinity of Cra toward P_{ptsG} . The experimental condition might be optimized still more to clear the interaction between c-di-GMP and Cra. But, for now, we speculate that the binding affinities of Cra differ between promoters and that, as a result, the effects of c-di-GMP on the binding affinity of Cra vary between promoters. The result has been added as Fig. S4 with the interpretation (line 322 - 327).

9. One weakness of the paper is that all the conclusions are based on gene expression results. It would strengthen the paper to demonstrate a functional consequence to the demonstrated cdG gene regulation. Are there specific growth conditions or substrates that can be used to demonstrate that cdG is decreasing PTS activity while increasing gluconeogenesis?

Response: We totally agree with the reviewer's suggestion. In order to verify that c-di-GMP downregulates the expression of three PTSs, PTS^{Glc} , PTS^{Man} , and PTS^{Fru} , and subsequently slows sugar transport by these PTSs, the amount of extracellular glucose in the culture broth was measured. A wild-type *Salmonella* consumed 50% glucose at 3 h, when the genes of PTS^{Glc} and PTS^{Man} were highly activated.

However, *Salmonella* producing c-di-GMP consumed only 24% glucose for the same time, which was presumably due to the downregulation of PTS^{Glc} and PTS^{Man}. The result has been added in the revised manuscript (line 137 - 139) as Fig. 2C. The method has been also supplemented with the additional glucose quantification assay (line 426 - 429).

10. The authors use standard T-tests to analyze statistical differences in large datasets, but they should really be using ANOVAs.

Response: According to the suggestion, we used one-way analysis of variance (ANOVA) with Tukey's post hoc test in the revised figures, including Fig. 4, Fig. 5, Fig. 6, Fig. 7, and Fig. S2. The method describing statistical analysis has been corrected accordingly (line 461 - 464).

Reviewer #2 (Comments for the Author):

This manuscript shows a new role for c-di-GMP, which is the modulation of carbon flux. The authors analyzed the effect of the overproduction of c-di-GMP in *S. Typhimurium*, by overexpressing the diguanylate cyclase AdrA, and found that it decreased the expression of genes encoding components of phosphoenolpyruvate (PEP):carbohydrate phosphotransferase systems (PTSs) for the uptake of glucose, mannose and fructose. Furthermore, c-di-GMP was shown to act through the regulators SrgS, M1c and Cra to control the three PTSs, being Cra the major factor for this c-di-GMP-mediated regulation. The authors demonstrate that CrA binds to genes of the three PTSs. Additionally, they found that c-di-GMP-Cra also regulates the expression of genes for gluconeogenesis but positively. Thus, this manuscript supports

that c-di-GMP rewires carbon flux from glycolysis to gluconeogenesis to favor biofilm formation, which requires the end products of gluconeogenesis.

The manuscript is well written, and conclusions are completely supported by data.

Response: We appreciate the reviewer's favorable evaluation. The detailed responses to each comment are listed below.

Comments.

What about deletion of *adrA*? Does it affect the expression of genes for carbon flow?

Response: We also considered to delete *adrA* to investigate the roles of c-di-GMP. However, the deletion of *adrA* did not influence the transcription of genes responsible for the production of curli and cellulose, which are primarily regulated by c-di-GMP. We speculated that other diguanylate cyclases except AdrA might have supplemented the absence of *adrA*. Due to multiple alternatives to AdrA, we decided to overexpress c-di-GMP, instead, to explore its role in *Salmonella* physiology. Although we have not examined the transcription of genes associated with PTSs and gluconeogenesis, it's unlikely that the deletion of *adrA* will change their expression.

It would be good to mention in Discussion that results from EMSAs (number or DNA/protein complexes) agree with the number of predicted Cra-binding sites on *fruB*, *pstG* and *manX*.

Response: As suggested by the reviewer, we asserted in Discussion that "We found that Cra

could directly bind to the promoters of *fruBKA*, *ptsG*, and *manXYZ* containing the consensus Cra boxes and repress their transcription”, indicating that the EMAS results were consistent with the predicted Cra-binding sites (line 310 - 312).

Do the genes for gluconeogenesis also harbor the CsrA-binding consensus sequence?

Response: The Csr system composed of CsrA (regulator), CsrB/CsrC (noncoding RNAs), and CsrD (regulator for CsrB/CsrD) is known to positively regulate glycolysis while inhibiting gluconeogenesis. The consensus binding motif of CsrA (ARGGAN, where GGA is mandatory) has been predicted in multiple genes associated with gluconeogenesis as well as glycolysis (Liu B. et al. 2021. *Microorganisms* 9(11): 2383). However, we don't think that the reviewer questioned the CsrA-binding sites in gluconeogenesis-relevant genes, because we've not tested and mentioned Csr system in the manuscript. If the reviewer intended Cra instead of CsrA, lots of studies have suggested the binding sites of Cra in the genes of gluconeogenesis and glycolysis (Kim D. et al. 2018. *Nucleic Acids Res.* 46(6): 2901-2917).

Minor comments.

Line 218. pCra instead pFruR

Response: We appreciate the thoughtful correction. The expression has been corrected to pCra throughout the text (line 220 - 222; 224 - 225).

Line 226. The bands from EMSAs were seen by staining with ethidium bromide, not by using an anti-His antibody.

Response: We appreciate the thorough review. The wording has been corrected according to the suggestion (line 232 – 233).

We feel that the reviewers' advice has greatly improved the manuscript. Thank you for handling the manuscript.

Yours sincerely,

Hyunjin Yoon, Ph.D

Department of Molecular Science and Technology,

Department of Applied Chemistry and Biological Engineering,

Ajou University, Suwon, 16499, Republic of Korea

+82-031-219-2450

yooh@ajou.ac.kr

January 16, 2023

Prof. Hyunjin Yoon
Ajou University
Suwon
Korea (South), Republic of

Re: Spectrum03685-22R1 (Cyclic di-GMP modulates a metabolic flux for carbon utilization in *Salmonella enterica* serovar Typhimurium)

Dear Prof. Hyunjin Yoon:

Your manuscript has been accepted, and I am forwarding it to the ASM Journals Department for publication. You will be notified when your proofs are ready to be viewed.

Sincerely,

Cristina Solano
Editor, Microbiology Spectrum
